# Med42 - Evaluating Fine-Tuning Strategies for Medical LLMs: Full-Parameter vs. Parameter-Efficient Approaches

**Clément Christophe[1], Praveen K Kanithi[1], Prateek Munjal[1], Tathagata Raha[1], Nasir Hayat[1], Ronnie Rajan[1], Ahmed Al-Mahrooqi[1], Avani Gupta[1], Muhammad Umar Salman[1], Boulbaba Ben Amor[2], Marco AF Pimentel[1], Shadab Khan[1]**

[1]G42 Healthcare [2]Core42, Abu Dhabi, UAE
{first name}.{last name}@g42healthcare.ai

## Abstract

This study presents a comprehensive analysis and comparison of two predominant fine-tuning methodologies – full-parameter fine-tuning and parameter-efficient tuning – within the context of medical Large Language Models (LLMs). We developed and refined a series of LLMs, based on the Llama-2 architecture, specifically designed to enhance medical knowledge retrieval, reasoning, and question answering capabilities. Our experiments systematically evaluate the effectiveness of these tuning strategies across various well-known medical benchmarks. Notably, our medical LLM Med42 showed an accuracy level of 72% on the US Medical Licensing Examination (USMLE) datasets, setting a new standard in performance for openly available medical LLMs. Through this comparative analysis, we aim to identify the most effective and efficient method for fine-tuning LLMs in the medical domain, thereby contributing significantly to the advancement of AI-driven healthcare applications.

## Introduction

In recent years, large language models (LLMs) have emerged as transformative tools, demonstrating remarkable proficiency in natural language understanding across a plethora of general-purpose applications, and sparking the interest in their use within specialized fields, particularly in the medical sector (Brown et al. 2020; Zhao et al. 2023; Zhu et al. 2023; Laskar et al. 2023). Notably, these models, such as OpenAI's GPT-3.5 and GPT-4 (OpenAI 2023), Google's BARD (Driess et al. 2023) as well as various other non-proprietary models, while initially developed for general natural language processing tasks, have been evolving and adapting to address the nuanced and complex language of the medical domain (Nori et al. 2023; Singhal et al. 2022).

To enhance the efficacy of LLMs for medical applications, researchers have recognized the importance of training and/or fine-tuning these models on large, in-domain datasets (Peng et al. 2023; Toma et al. 2023; Zhou et al. 2023). The utilization of such datasets allows for nuanced understanding of both natural language and domain-specific terminologies, making them adept at interpreting and generating text that aligns with the intricacies of medical language. Albeit concerns about hallucinations and fabrications, biases and

knowledge gaps, and risks about data privacy and ethics (Thirunavukarasu et al. 2023; Li et al. 2023a), this specialized capability enables LLMs to be potentially employed in various healthcare applications. These include aiding diagnostic processes by analyzing and summarizing patient history and symptoms (Souza et al. 2023; Hirosawa et al. 2023), interpreting medical literature and research papers for easier comprehension by healthcare professionals (Cascella et al. 2023; Bagde et al. 2023), generating patient education materials tailored to individual needs (Ali et al. 2023; Miner, Laranjo, and Kocaballi 2020), and assisting in the development and consultation of clinical guidelines and decision support systems (Wang et al. 2023; Hamed, Eid, and Alberry 2023).

To this end, we focus on the development of a medical LLM, by comparing and analyzing two predominant fine-tuning methodologies: full-parameter fine-tuning and parameter-efficient tuning. Full-parameter fine-tuning is a comprehensive approach that involves adjusting all parameters of a pre-trained model, which demands substantial computational resources and time (Ding et al. 2023). In contrast, parameter-efficient tuning methods, such as Adapters (Houlsby et al. 2019; Lin, Madotto, and Fung 2020), Low-Rank Adaptation (LoRA) (Hu et al. 2022), and Prompt-tuning (P-tuning), (Li and Liang 2021; Lester, Al-Rfou, and Constant 2021; Liu et al. 2023) offer a more resource-efficient alternative by modifying a smaller subset of the model's parameters. This study presents a detailed comparison of these methods, specifically within the context of medical LLMs. Our investigation includes experiments to assess the effectiveness of these tuning strategies, with a particular focus on the emerging LoRA technique. Through this comparative analysis, our objective is to identify the effective and efficient method for fine-tuning LLMs in the medical domain, ultimately contributing to the advancement of AI-driven healthcare applications. We are also releasing our most performant model Med42 on HuggingFace[1].

## Methods

In this section, we elaborate on the dataset comprising our study, detailing its characteristics and the rationale for its selection. We outline the specific methodologies implemented

---

[1]https://huggingface.co/m42-health/med42-70b

for fine-tuning our large medical language model, including a comparison of full-parameter tuning and parameter-efficient techniques such as Low-Rank Adaptation (LoRA). Furthermore, we provide an exhaustive list of hyperparameters and configurations employed during our experiments, aiming to offer a transparent and replicable framework for subsequent research endeavors in this domain. This section aims to provide a comprehensive understanding of the technical aspects and decision-making processes underpinning our model's development and evaluation.

## Training Dataset

Our instruction-tuning dataset is a combination of multiple open datasets, primarily focused on medical question-answering data. It includes an extensive collection from medical forums, notably those within the Stack Exchange network, which are rich with expert discussions, patient inquiries, and specialist responses. Additionally, we incorporated selected sections from general domain datasets, meticulously extracting and integrating segments specifically related to medical topics. This composite approach ensures a diverse and robust dataset, encompassing a wide range of medical subfields and contexts, providing a comprehensive foundation for training our model to understand and generate medically-relevant content accurately. Further details about the training dataset are described in Appendix .

In order to make our model learn from instructions effectively, we employed a structured instruction format using the keywords <|system|>, <|prompter|>, and <|assistant|>. This format has been designed to teach the model the relationship between a given command and its appropriate output. By encapsulating the input under <|prompter|>, the intended system operation under <|system|>, and the expected output under <|assistant|>, we created a clear, directive framework that aids the model in understanding and executing tasks based on instructions.

## Modelling

**Models.** In this study, we built on the Llama-2 (Touvron et al. 2023b) family of models as the foundational architecture for fine-tuning. We specifically focused on the 7 billion (7B) and 70 billion (70B) parameter versions of these models. These versions were selected for their robust pre-training and scalability, allowing us to explore the impact of model size on performance in medical domain-specific tasks. Also, Llama-2 model comes with an open license, allowing for greater flexibility for adaptation and use in our research.

**LoRA.** Low-Rank Adaptation (LoRA) is a parameter-efficient training technique that targets the adaptation of pre-trained language models without the need for full model fine-tuning. Instead of updating all the parameters, LoRA focuses on a subset of the Transformer architecture. It introduces trainable low-rank matrices while keeping the pre-trained weights fixed. These matrices capture the essential changes needed to adapt the model to new tasks, effectively

reducing the number of trainable parameters and thus computational costs.

The selection of layers to which LoRA is applied constitutes a hyperparameter that requires careful tuning to optimize model performance. While it is common to see LoRA applied only to attention layers $v\_proj$, $q\_proj$, $k\_proj$ and $o\_proj$ or only $gate\_proj$, $down\_proj$, and $up\_proj$ as in (He et al. 2021; Lee, Hunter, and Ruiz 2023), we achieved the best performance by applying it to every linear layer as in (Dettmers et al. 2023). With these settings, the number of trainable parameters goes from 7 and 70 billion to 20 and 104 million, respectively. Details about the computational setup are available in Appendix 2.

**Mask loss.** In our methodology, every sample is composed of three elements: a system prompt, a user prompt, and a corresponding response. To optimize the use of the model's available context length, we concatenate these samples across the entire training dataset. The training approach is autoregressive, focusing the backpropagation of loss exclusively on the tokens forming the responses. Consequently, this training strategy ensures that the model predominantly learns to generate answers, rather than the prompts.

**Hyperparameters.** We train using the AdamW optimizer (Loshchilov and Hutter 2017), with $\beta_1 = 0.9$ and $\beta_2 = 0.95$. We use a cosine learning rate schedule, with a linear warmup of 100 steps, and decay final learning rate to 10% of its peak. We use a weight decay of 0.1 and gradient clipping of 1.0. For full-parameter fine-tuning, we trained the model for 3 epochs with a peak learning rate of $5e^{-5}$. For LoRA fine-tuning, we trained for 8 epochs with a peak learning rate of $1e^{-4}$ and $\alpha = 16$ and $r = 8$. To speed up the training, we packed all of our fine-tuning data into chunks of 4,096 tokens.

## Model evaluation

To assess the performance of the fine-tuned language models, following previous works (Singhal et al. 2023; Chen et al. 2023; Toma et al. 2023), we used Eleuther AI's evaluation harness framework (Gao et al. 2021) to compute their zero-shot performance across various commonly-used medical benchmarks. These contain medical exam questions and research datasets with multiple-choice answers, and include MedQA, HeadQA, MedMCQA, PubMedQA, MMLU clinical topics, and both self-assessment and sample exams from the United States Medical Licensing Examination (USMLE). All datasets are in the English language and all questions containing images were excluded. We describe these datasets in more detail below.

**MedQA.** The dataset consists of multiple-choice (4 or 5) questions that resemble USMLE questions (from the National Medical Board Examination in the USA) and was originally designed for addressing medical problems (Jin et al. 2020).

**HeadQA.** This multiple-choice question-answering dataset which is sourced from exams to access a specialized position in the Spanish healthcare system (Vilares and

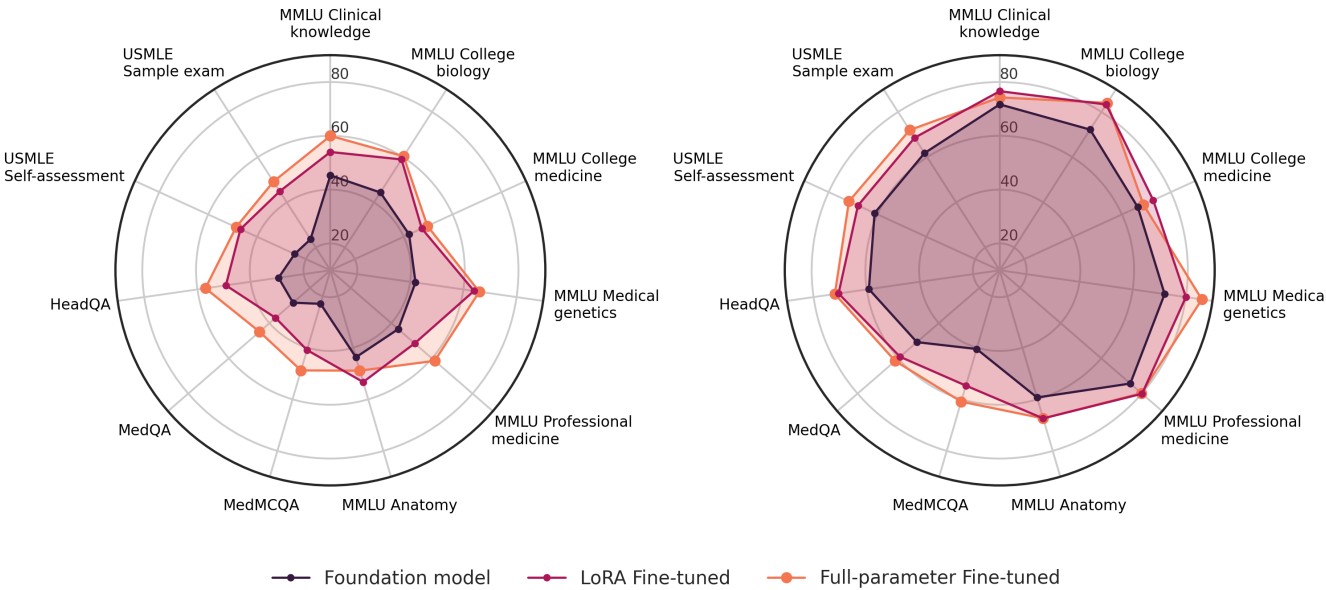

Figure 1: Performance of 7-billion (left) and 70-billion (right) parameter models on various medical-related benchmark datasets (in zero-shot setting). Performance results (accuracy) are displayed in % for the base and fine-tuned models.

Gómez-Rodríguez 2019). Only the English subset has been included in our evaluation.

**MedMCQA.** A large-scale multiple-choice questions dataset (4-choices) from the Indian medical entrance examinations (Pal, Umapathi, and Sankarasubbu 2022), covering 21 medical subjects and more than 2,000 healthcare topics. We report the performance of our models on the validation set (given that the available test dataset does not contain answers to the questions), which has been excluded from our training (fine-tuning) dataset.

**PubMedQA.** The task of PubMedQA is to answer research questions with yes/no/maybe using abstracts from medical scientific literature (Jin et al. 2019); i.e., given an abstract and a related question, the task is to provide a short answer of yes, no or maybe.

**MMLU clinical topics.** The widely-used Measuring Multitask Language Understanding (MMLU) benchmark (Hendrycks et al. 2021) aimed to introduce a comprehensive assessment of LLMs across 57 subjects. For our evaluation, we selected clinical topics covering clinical knowledge, college biology, college medicine, medical genetics, professional medicine and anatomy.

**USMLE sample exam and self-assessment.** These are two sets of official practice materials for the United States Medical Licensing Examination (USMLE), which is an examination program that contains three steps for assessing medical competency (Nori et al. 2023; Han et al. 2023).

We also address a growing concern in the field of LLM fine-tuning: the inadvertent inclusion of similar or identical samples in both training and evaluation datasets. To address this, we implemented a "decontamination pipeline", which is designed to scrutinize the evaluation dataset and flag any examples that have a significant resemblance to those found in our instruction-tuning dataset. These flagged examples are regarded as "contaminated samples". To ensure the integrity of our evaluation, we also show the performance metrics calculated after the removal of these contaminated samples from the evaluation datasets. Detailed information about the decontamination process can be found in Appendix.

## Results

The performance of the fine-tuned models across the different benchmark datasets is represented in Figure 1, showing the accuracy values obtained for both 7B and 70B-parameter models. The results show the superiority of the fine-tuned models over their corresponding base models across all medical benchmark datasets. Notably, our analysis reveals that full-parameter fine-tuning outperforms the parameter-efficient fine-tuning approach, LoRA, in the majority of these datasets.

The comparative results from our experiments, illustrating the performance of our best fine-tuned models (70B-parameter models) against that of other models, are detailed in Table 1. Changes in the accuracy scores after the removal of contaminated examples are shown in Figure 2.

## Discussion

The findings of this study underscore the efficacy of comprehensive fine-tuning strategies in enhancing the performance of language models. Importantly, we report domain-adapted fine-tuned medical LLMs that demonstrate high-level medical reasoning and improved domain-specific benchmark performance, particularly, in medical complex tasks such as

| Dataset | PE-FT | FP-FT (Med42) | ClinCamel[1] | MediTron[2†] | GPT-3.5[3] | GPT-4[3] | MedPaLM-2[4†] |
|---|---|---|---|---|---|---|---|
| **MMLU** (average) | **76.7** | **76.7** | 69.8 | 71.5 | 66.6 | 87.0 | **89.4** |
| Clinical knowledge | **76.6** | 74.3 | 69.8 | - | 69.8 | 86.0 | **88.3** |
| College biology | 83.3 | **84.0** | 79.2 | - | 72.2 | **95.1** | 94.4 |
| College medicine | **72.8** | 68.8 | 67.0 | - | 61.3 | 76.9 | **80.9** |
| Medical genetics | 80.0 | **86.0** | 69.0 | - | 70.0 | **91.0** | 90.0 |
| Professional medicine | **80.1** | 79.8 | 71.3 | - | 70.2 | 93.0 | **95.2** |
| Anatomy | **67.4** | **67.4** | 62.2 | - | 56.3 | **80.0** | 77.8 |
| **HeadQA** | 70.6 | **72.0** | - | - | - | - | - |
| **MedMCQA** | 54.7 | **60.9** | 47.0 | 53.3 | 50.1 | 69.5 | **71.3** |
| **MedQA** | 59.1 | **61.5** | 53.4 | 52.0 | 50.8 | 78.9 | **79.7** |
| **PubMedQA** | 75.8 | 76.8 | 74.3 | **79.8** | 71.6 | 75.2 | **79.2** |
| **USMLE** (average) | 68.3 | **71.9** | - | - | 53.1 | **84.1** | - |
| Self-assessment | 68.0 | **71.7** | - | - | 49.1 | **83.8** | - |
| Sample exam | 68.6 | **72.0** | 54.3 | - | 56.9 | **84.3** | - |

Table 1: Zero-shot performance comparison between both parameter-efficient (PE-FT) and full-parameter (FP-FT) fine-tuned (Llama2 70-B) models with other published models across the various medical benchmark datasets. [1]Clinical Camel (Toma et al. 2023); [2]MediTron (Chen et al. 2023); [3]GPT-3.5 and GPT-4 (Nori et al. 2023); [4]MedPaLM-2 (Singhal et al. 2023). [†]We note that zero-shot performance is not reported for these models; few-shot results are shown.

USMLE-based questions.

Overall, full-parameter fine-tuning achieved better performance than parameter-efficient fine-tuning in medical tasks (Figure 1). However, it is noteworthy that parameter-efficient fine-tuning methods, such as LoRA, yield results that are remarkably close to those achieved by full-parameter fine-tuning, consistent with findings in other studies (Fu et al. 2023; Liao, Meng, and Monz 2023). These findings suggest that parameter-efficient approaches can be viable alternatives, particularly in scenarios where computational resources are limited.

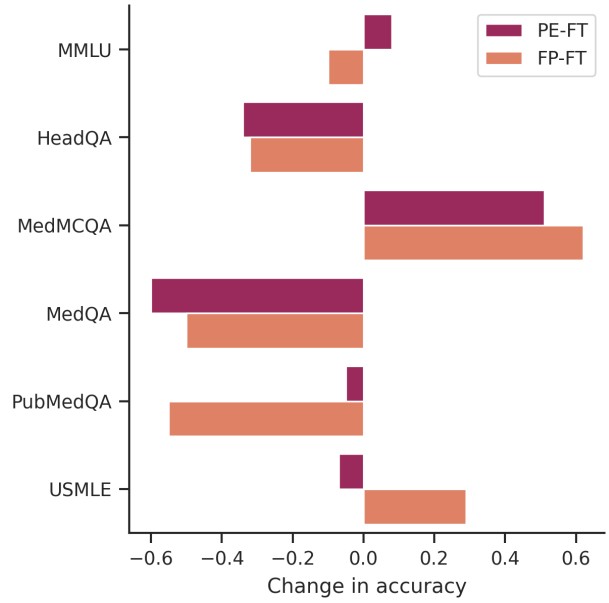

Figure 2: Accuracy change after decontamination for both (70b) fine-tuned models (shown in %).

A critical aspect of our study was the thorough examination of potential test set contamination. We analyzed whether our evaluation datasets contained examples that were either identical or strikingly similar to those in the training set, and re-evaluated our models on the "decontaminated" dataset. We observed that a small number of samples were deemed to be "contaminated" (Table A2), which resulted in very small changes in the accuracy scores for the two larger fine-tuned models across the benchmark datasets (Figure 2). This process ensures the robustness and integrity of our analysis, affirming the reliability of our findings and the trained models.

Moreover, our study encompassed a comparative analysis of the performance of fine-tuned models against other LLMs, including those commercially available. This comparison provides a more comprehensive view of the standing of our fine-tuned models in the current landscape of openly available LLMs, particularly in medical applications.

This research also underscores the importance of creating a large and well-structured instruction fine-tuning dataset. As instruct fine-tuning of open-source LLMs becomes a *de facto* standard practice, our results show that our model exhibits superior performance compared to established names like ClinicalCamel (Toma et al. 2023), MediTron (Chen et al. 2023) and GatorTronGPT (Peng et al. 2023). Our approach involved minimal experimentation with complex prompt engineering; however, we believe there are additional opportunities to enhance the model's response quality and accuracy. These opportunities could be explored in future research to achieve greater advancements.

While our results regarding the model's performance on medical question-answering benchmarks are promising, they also highlight areas for future exploration. Our study serves as a stepping stone, indicating the potential of these models in diverse applications. However, further research is necessary to fully understand and demonstrate the utility of

these models in other practical use-cases. Such investigations are crucial for advancing the field and realizing the full potential of fine-tuned LLMs in a variety of domains.

## Acknowledgments

This work was supported in part by the Cerebras Team.

## Ethical Considerations and Reproducibility

In this work, we underscore the critical importance of ethical considerations and reproducibility in the development of our large language model. Firstly, our model is released under an open license, ensuring public accessibility and fostering a culture of transparency and collaboration within the research community. This approach not only democratizes access to advanced technology but also invites scrutiny and diverse perspectives, which are vital for ethical development. Additionally, the datasets and frameworks used in our model's evaluation are freely available, promoting reproducibility and allowing independent verification of our findings. This transparency in data and methodologies is essential to maintain scientific integrity and foster trust in AI research. Furthermore, we recognize and openly acknowledge the limitations of our model, particularly its readiness for use in clinical practice. We emphasize that while our model shows promising results, it should not yet be used as a decision-making tool in clinical environments. There is the potential for generating incorrect or harmful information and the risk of perpetuating biases in training data. This acknowledgment stems from our commitment to responsible AI development, where the safety and well-being of end-users are paramount. Ongoing human evaluation efforts are focused on rigorously assessing biases, fairness, and safety through red-teaming the model to ensure its robustness and reliability in clinical settings. By addressing these ethical concerns and emphasizing reproducibility, we aim to contribute positively to the field of AI, ensuring that advancements are made with a keen awareness of their societal impacts and limitations.

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

# Appendix: Supplementary Materials

## A.1 Related work

Most contemporary LLMs predominantly employ a decoder-only transformer architecture such as GPT-3 (Brown et al. 2020) for generating human-like text. Such models, typically with billions or hundreds of billions of parameters, are trained using large amounts of diverse textual data, leading to the emergence of significant generative capabilities in various tasks. Particularly, the landscape of LLMs has predominantly been shaped by models initially trained for general-purpose tasks, with some later being adapted for specialized domains like healthcare.

Notable examples in this realm include GPT-3.5 and GPT-4, developed by OpenAI, which have demonstrated impressive capabilities across a wide range of tasks, including high performance on various medical benchmarks (Nori et al. 2023). Similarly, Singhal et al. demonstrated state-of-the-art performance of Google's Med-PaLM and Med-PaLM 2 (Singhal et al. 2022, 2023). Albeit using a general-purpose trained model (PaLM), the authors applied an ensemble refinement, computational heavy prompting strategy which showed increased performance on medical examination and other medical-related question-answering datasets as well as encouraging results on human evaluation experiments. However, it is important to note that intricate details concerning the training methodologies and weight parameters for such models remain undisclosed.

There has been an interest in tailoring LLMs for specific biomedical domains by pre-training the model using domain-specific datasets. Models such as GatorTron and its successor, MedGatorTron, exemplify this approach (Yang et al. 2022; Peng et al. 2023). These models leverage large corpora of medical data, including electronic health records (EHRs), clinical notes, and medical literature to build a robust foundational understanding of medical terminologies, concepts, and patient-care scenarios. This pre-training on domain-specific data sets the stage for more effective and nuanced applications in the medical field. PMC-LLaMA is another recently-released model, which was initially pre-trained on a biomedical/clinical corpus, and subsequently trained on an instruction dataset primarily containing medical question answering and reasoning tasks (Wu et al. 2023).

Another significant trend in the development of healthcare-oriented LLMs has been the utilization of medical-related instruction and dialogue datasets for fine-tuning. This "alignment" approach involves fine-tuning a pre-trained model on a collection of instruction-following samples, to effectively improve the zero-shot and few-shot generalization abilities of LLMs. Chat-Doctor (Li et al. 2023b), for example, is a medical LLM fine-tuned (using full-parameter fine-tuning) on a Large Language Model Meta-AI (LLaMA) (Touvron et al. 2023a) using a dataset that contains online conversations between physicians and patients, and compared favourably to GPT-3.5 when evaluated using a number of medical queries. Similarly, Han et al. propose MedAlpaca by fine-tuning LLaMA using a collection of over 160,000 medical NLP tasks reformatted in instruction tuning formats as well as a crawl of various internet resources (Han et al. 2023). The authors developed a parameter-efficient variant (using LoRA) which was shown to underperform (when compared to the corresponding full-parameter fine-tuned variant) on the United States Medical Licensing Examination (USMLE) self-assessment dataset. Other more recent models, which were developed concurrently with those presented in this manuscript, include models based on LLama-2 (Touvron et al. 2023b), such as Clinical Camel (Toma et al. 2023) and MediTron (Chen et al. 2023). In the latter, we note that model fine-tuning was conducted using the training set of each individual benchmark that the model was evaluated, and that fine-tuning was preceded by a phase of continuous pre-training using biomedical data sources such as PubMed Central and PubMed open-access research papers.

Our study builds upon the approaches described above, extending the scope by conducting a more thorough comparison of different fine-tuning methods of LLMs within the medical domain. This work not only delves into the nuances of these tuning strategies, including the LoRA technique but also rigorously evaluates their effectiveness across multiple benchmarks. This comparative analysis is pivotal in pinpointing the most effective and efficient fine-tuning approach for LLMs in the medical sector, thereby making a significant contribution to the evolution of AI in healthcare.

## A.2 Computational Setup

For the parameter-efficient fine-tuning and evaluation processes, we utilized 8 nodes equipped with 16 NVIDIA V100 GPUs each. This configuration facilitated efficient experimentation and analysis of the fine-tuning methods under consideration. In contrast, the full-parameter fine-tuning experiments were conducted on the Condor Galaxy 1 (CG-1) supercomputer, provided by Cerebras. The CG-1 supercomputer offered the necessary computational power and infrastructure to handle the extensive demands of full-parameter fine-tuning processes.

## A.3 Training dataset

In the construction of our training dataset, we selected a range of datasets specifically related to medical and biomedical fields, ensuring relevance and applicability to our study's focus. Recognizing the extensive size of some of these datasets, we employed a strategic sampling approach to extract representative subsets, thereby maintaining a comprehensive, yet manageable dataset size. Additionally, to provide a broader linguistic context and enhance the model's generalizability, we incorporated a dataset from a general domain. This subset was carefully chosen so that the general domain data constituted 40% of the final training dataset. This hybrid dataset composition was designed to optimize the model's performance across both specialized medical content applications and more general linguistic tasks.

Table A1 provides a detailed overview of the various subsets of data included in our study, along with the number of samples contained in each subset.

| Dataset | # Samples | Mixture ratio (%) |
|---|---|---|
| **Medical domain** | | |
| MedMCQA (Pal, Umapathi, and Sankarasubbu 2022) | 180,462 | 25.54 |
| Medical Flashcards (Han et al. 2023) | 30,106 | 4.26 |
| StackExchange[†] (Lambert et al. 2023) | 64,246 | 9.09 |
| MedQA (USMLE) (Jin et al. 2020) | 11,290 | 1.60 |
| CORD-19 (Wang et al. 2020) | 17,721 | 2.51 |
| PubMedQA (Jin et al. 2019) | 499 | 0.07 |
| HeadQA[‡] (Vilares and Gómez-Rodríguez 2019) | 2,657 | 0.38 |
| MediQA (Han et al. 2023) | 1,950 | 0.28 |
| PubMed Health Advice (Han et al. 2023) | 7,694 | 1.09 |
| PubMed Causal (Han et al. 2023) | 2,169 | 0.31 |
| OpenGPT | 66,026 | 9.34 |
| MedQUAD (Ben Abacha and Demner-Fushman 2019) | 14,553 | 2.06 |
| MMLU[$] (Hendrycks et al. 2021) | 244 | 0.03 |
| Niv2* (Wang et al. 2022) | 11,447 | 1.62 |
| Total | 411,064 | 58.17 |
| **General domain** | | |
| OpenOrca T0 (Lian et al. 2023; Sanh et al. 2022) | 110,905 | 15.69 |
| OpenOrca Flan (Lian et al. 2023; Longpre et al. 2023) | 110,854 | 15.69 |
| OpenOrca CoT (Lian et al. 2023; Wei et al. 2022) | 73,890 | 10.46 |
| Total | 295,649 | 41.83 |

[†] The following categories were included: "academia", "bioinformatics", "biology", "cogsci", "fitness", "health".

[‡] Only samples in English were used.

[$] The following subjects were included: "anatomy", "clinical knowledge", "college medicine", "medical genetics", "professional medicine", "college biology", "high-school biology", "professional psychology", "high-school psychology", "human sexuality", "human aging", "nutrition", and "virology".

* Samples from 47 tasks (from over 1,000 tasks) related to science, healthcare and medicine were included.

A.1: Summary of subsets of the data used for fine-tuning the models. Note that medical-domain data correspond to approximately 60% of the entire dataset.

## A.4 Decontamination analysis

To address the unintentional inclusion of similar or identical samples in both the training and evaluation datasets, we implemented a "decontamination pipeline". As in (Lee, Hunter, and Ruiz 2023), we compute the cosine similarity, as measured by SentenceTransformers embeddings, between our instruction-tuning dataset and each sample in the evaluation dataset. We deem a sample as "contaminated" if the cosine similarity exceeds 0.8. Our analysis reveals that most duplicates are simply reworded versions of the same question. However, some are extensive use-case questions that, despite not being directly the same, contain many identical words.

Table A.2 details the number of samples that were deemed to be contaminated. We observe that less than 2% of the samples in our evaluation dataset appear in our instruction-tuning dataset. We also observe that the majority of these duplicates are differently phrased questions about similar diseases or similar clinical concepts. In the case of longer use-case questions, they are deemed similar due to a high overlap in wording, despite having different final questions. Examples of contaminated examples are shown in Figure A.1.

In the interest of transparency, we re-calculated our evaluation metrics after excluding these potentially contaminated samples (Figure 2, Table A3). The results demonstrate no substantial deviation from our original findings (Table 1) after removing the "contaminated" examples. By identifying and segregating these samples, we aimed to ensure a more accurate and reliable assessment of the models' performance.

| Dataset | Samples | Contaminated (%) |
|---|---|---|
| MMLU (total) | 1,089 | 17 (1.6) |
| Clinical Knowledge | 265 | 1 (0.4) |
| College Biology | 144 | 1 (0.7) |
| College Medicine | 173 | 1 (0.6) |
| Medical Genetics | 100 | 6 (6.0) |
| Professional Medicine | 272 | 7 (2.6) |
| Anatomy | 135 | 1 (0.7) |
| MedMCQA | 4183 | 65 (1.6) |
| MedQA | 1273 | 73 (5.7) |
| HeadQA | 2742 | 62 (2.3) |
| PubmedQA | 500 | 2 (0.4) |
| USMLE (total) | 650 | 22 (3.4) |
| Self-Assessment | 325 | 11 (3.4) |
| Sample Exam | 325 | 11 (3.4) |
| **Total** | 11904 | 280 (2.4) |

A.2: Results of the de-duplication pipeline over evaluation datasets.

| Dataset | PE-FT | ± | FP-FT | ± |
|---|---|---|---|---|
| **MMLU** (average) | 76.8 | (+0.1) | 76.6 | (-0.1) |
| Clinical knowledge | 76.9 | (+0.3) | 74.2 | (-0.1) |
| College biology | 83.2 | (-0.1) | 83.9 | (-0.1) |
| College medicine | 73.2 | (+0.4) | 69.2 | (+0.3) |
| Medical genetics | 79.8 | (-0.2) | 85.1 | (-0.9) |
| Professional medicine | 80.4 | (+0.3) | 80.0 | (+0.2) |
| Anatomy | 67.2 | (-0.2) | 67.2 | (-0.2) |
| **HeadQA** | 70.3 | (-0.3) | 71.7 | (-0.3) |
| **MedMCQA** | 55.2 | (+0.5) | 61.5 | (+0.6) |
| **MedQA** | 58.5 | (-0.6) | 61.0 | (-0.5) |
| **PubMedQA** | 75.7 | (-0.1) | 76.3 | (-0.5) |
| **USMLE** (average) | 68.2 | (-0.1) | 72.1 | (+0.2) |
| Self-assessment | 67.7 | (-0.3) | 72.2 | (+0.5) |
| Sample exam | 68.7 | (+0.1) | 72.0 | (0.0) |

A.3: Zero-shot accuracy for our 70B model after removing contaminated samples from evaluation datasets. Comparison with full results presented in Table 1.

**Train**

**Evaluation**

**In a patient with head injury, eye opening is seen with painful stimulus, localizes the pain and there is inappropriate verbal response. What would be score on Glasgow coma scale?**
**a. 8**
**b. 9**
**c. 10**
**d. 11**

c. 10

**Before a patient with traumatic brain injury, what score on the Glasgow Coma Scale does it show if we observe that it emits inappropriate words, opens its eyes when speaking to it and makes a withdrawal response when applying a painful stimulus?**
**a. 11**
**b. 10**
**c. 9**
**d. 8**

b. 10

—————-

**Uvula vesicae seen in bladder is formed from the following structure ?**
**a. Lateral lobe of prostate**
**b. Median lobe of prostate**
**c. Anterior lobe of prostate**
**d. Posterior lobe of prostate**

b. Median lobe of prostate

**Uvula vesicae is produced by which prostate lobe?**
**a. Anterior lobe**
**b. Post lobe**
**c. Median lobe**
**d. Lateral lobe**

c. Median lobe

A.1: Two examples of contaminated samples from our instruction-tuning (left) and evaluation datasets.