# OpenReview forum: "Med42 - Evaluating Fine-Tuning Strategies for Medical LLMs: Full-Parameter vs. Parameter-Efficient Approaches"
_AAAI.org/2024/Spring_Symposium_Series/Clinical_FMs — AAAI 2024 SSS on Clinical FMs_

### Official Review · Reviewer_txoL · 2024-02-19
**Comparison of LoRA and Full-Parameter Fine-Tuning LLMs for Medical Q&A**

**Rating:** 8
**Confidence:** 3

**Review:**

## Summary
This paper investigates the effectiveness of using LoRA fine-tuning compared to full-parameter fine-tuning LLMs for adaptation to the healthcare domain. Authors report results for both smaller 7B Llama-2 as well as larger 70B Llama-2 fine-tuned models. They also compare against close-source state-of-the-art models like GPT-4 and Med-PaLM-2. The findings and described methods are a useful reference for healthcare machine learning practitioners who want to fine-tune a general-domain LLM for health-care tasks.

## Pros
* Evaluation is comprehensive across multiple datasets
* Evaluation is carefully done with decontamination pipeline
* Compares one of the most popular parameter efficient fine-tuning technique, LoRA, against full fine-tuning and state-of-the art models (GPT-4 and Med-PaLM) in healthcare domain
* Models are publicly released and available
* Datasets are publicly released and available
* Manuscript is well written and easy to follow

## Cons
* Methods section describes that LoRA may be applied to only attention layers vs. all layers. PE-FT results in Table 1 are for LoRA applied to all layers. It would be nice to also show the performance for LoRA applied to only attention layers since authors have mentioned this is a common approach. However, this is more of a "nice to have".

---

### Official Review · Reviewer_WxPy · 2024-02-21

**Rating:** 6
**Confidence:** 4

**Review:**

**Summary:**

The paper focuses on fine-tuning the 7B and 70B Llama-2 models on a dataset compiled from several open medical datasets, evaluating the performance gap between LoRA fine-tuning and full-parameter fine-tuning. Experiments conducted on several medical benchmarks led to good performance on some of the benchmarks, outperformed only by models trained at a larger scale (GPT-4) and models pre-trained on medical corpora (MedPaLM-2). To ensure a fair evaluation, the paper also introduces a decontamination pipeline to remove potential common samples between the training and the testing splits of the benchmarks.

**Strengths:**

1. The evaluation benchmark is thorough, encompassing a wide set of medical benchmarks, thus enabling a more in-depth analysis.

2. The dataset introduced by the authors seems fairly comprehensive and suitable for the clinical domain, and the performance obtained by the Llama models trained in the paper realistically substantiates this.

3. The focus on data decontamination for a fairer analysis by the authors is appreciated and makes their results more relevant.

4. The overall work presented in this paper is very relevant to the topic of the venue.

5. The elaborate (for a short paper) description of the hyperparameters to enable reproducibility is appreciated.

**Weaknesses:**

1. The theme of the paper revolves around parameter-efficient fine-tuning vs full-parameter fine-tuning. However, the claim that parameter-efficient fine-tuning achieves results close to full-parameter fine-tuning is a well-known research artifact. The authors themselves note that the results are in line with prior work for LoRA vs full-parameter fine-tuning in other domains. I would recommend the authors adjust the paper to better describe their main contributions towards the compilation of the training dataset from open medical sources and the data decontamination pipeline, with a lesser focus on parameter-efficient fine-tuning vs full-parameter fine-tuning.

2. I recommend the authors describe the instruction tuning methodology in greater detail in the main paper, if space permits, else in the appendix.

**Other recommendations:**

There is a typo in the caption for Table 1, where GPT-3.5 is incorrectly mentioned as GPT-3.4.

---

### Official Review · Reviewer_tRbZ · 2024-02-22
**Fine-tuning strategies evaluation**

**Rating:** 7
**Confidence:** 4

**Review:**

The authors evaluated the performance of different fine-tuning strategies on medical QA tasks. A variety of medical QA datasets were used in the evaluation. Full parameter fine-tuning vs. LoRA fine-tuning were used and compared. Zero-shot performance were used to evaluate the model performance.

Only two different size Llama-2 models were used as the base model. Some studies have shown that different base models may provide different level of improvement after fine-tuning, so it would be great to see the results from some other commonly available open source models such as mistral.

The results were not new and mostly expected. Similar studies have been conducted and the results of this study were generally consistent with some previous findings. The authors did include more QA datasets for the evaluation, which made the results more comprehensive.

---

### Official Review · Reviewer_VxwM · 2024-02-22

**Rating:** 7
**Confidence:** 4

**Review:**

Pros:
- The authors articulate a well-defined research question and address it with clarity and efficiency;
- There's a comprehensive evaluation conducted against various state-of-the-art large language models, both open-source and proprietary;
- The presentation of results is clear and straightforward.


Suggestion for improvement:
- It would be beneficial to explicitly indicate in Table 1 which backbone corresponds to PE-FT and FP-FT; I assume it's llama70b. Additionally, including a column for llama7b would give the complete picture.
- It would be insightful to detail the computational resources required, in terms of GPU hours and memory, for both PE and FP fine-tuning. Providing this comparison could offer a clearer understanding of the differences in resource intensity between the two methods.